# Solving Confirmation Time in Sharded Blockchain with PFQN

**Junting Wu** [1,2], **Haotian Du** [3], **Jin Chen** [1,2] and **Wei Ren** [1,2,*]

[1] College of Computer and Information Science, Southwest University, Chongqing 400715, China
[2] College of Software, Southwest University, Chongqing 400715, China
[3] College of Economics and Management, Southwest University, Chongqing 400715, China
[*] Correspondence: oicq@swu.edu.cn

**Abstract:** Sharding shows great potential for extending the efficiency of blockchains. The current challenge facing sharded blockchain technology lies in addressing the extended transaction confirmation times caused by isolated states between shards and unbalanced transaction allocation strategies. These factors contribute to an increase in cross-shard transactions and disproportionate shard workloads, ultimately resulting in indefinite confirmation delays for cross-shard transactions. A critical priority for sharded blockchain systems is to conduct a comprehensive qualitative analysis to better understand and mitigate the prolonged transaction confirmation times. We introduce a product-form queue network (PFQN) model to address the transaction confirmation time problem in sharded blockchains and incorporate a new confirmation queue to more accurately simulate the actual transaction confirmation process in the blockchain. In addition, we provide a detailed quantitative analysis of the relationship between the network load and consensus efficiency in sharded blockchains, offering a meaningful perspective for achieving robustness and efficiency in sharded blockchains. This research not only contributes to addressing the scalability issues in sharded blockchains but also offers a new perspective for future research directions.

**Keywords:** blockchain sharding; transaction confirmation time; cross-shard transactions; product-form queue network

## 1. Introduction

### 1.1. Research Background

Sharding is a promising approach for improving blockchain scalability by dividing the network into smaller partitions, each processing a subset of transactions (TXs), thereby enhancing transaction throughput. Sharded blockchains are constructed from three paradigms: network sharding, transaction sharding, and state sharding [1]. Network sharding forms the basis of other paradigms, creating partitions that handle distinct TX sets according to the transaction sharding policy. State sharding aims to distribute the blockchain's states evenly across all shards. They therefore split the work related to the network, computation, and storage across the blockchain systems. Currently, state sharding remains mostly theoretical. Representative sharding solutions include Elastico [2], Omniledger [3], RapidChain [4], and Monoxide [5], based on either Unspent Transaction Output (UTXO) or account/balance transaction models.

As a state replication machine, a blockchain requires cross-shard transactions to unify parts of the state across different state shards. Therefore, sharding technology has been introduced as a method for cross-shard transactions. A cross-shard transaction refers to a transaction (TX) involving accounts or UTXOs on multiple shards. Because cross-shard transactions require verification of the correctness of the shard state being sent, they are more complex and time-consuming than single-shard transactions. A study by Rapidchain pointed out that, as the number of shards increases, almost all TXs become cross-shard [4]. Therefore, reducing the number and delay of cross-shard transactions is key to improving the scalability of shard blockchains [6,7].

Transaction confirmation is mainly completed by consensus within shards and cross-shard consensus between shards. In sharded blockchains, transactions are first submitted to the relevant shard. Each shard has its independent verification process and input, so the transaction confirmation time varies depending on the shard's consensus mechanism. This paper discusses consensus mechanisms similar to Monoxide (Proof of Work, i.e., PoW, relay for cross-shard transaction).

### 1.2. Related Works

Compared to the extensive research on sharded blockchains, the literature exploring the application of queueing theory in analyzing blockchain characteristics is relatively limited. Still, some inspiring lines of research can be found in the literature.

In terms of applying queueing theory to blockchains, ref. [8] took significant steps forward. They used the GI/M/1 queue model with batch-service for single-chain system analysis. This work helped point out what is important in how blockchain systems perform, such as the average number of transactions and the duration of confirmation times. Then, ref. [9] integrates machine learning with queueing theory to enhance the understanding of confirmation times for transactions in single-chain systems. This research introduces a novel machine learning methodology for sorting transactions and applies queueing theory to assess delays.

In the context of PoW, ref. [10] established a model for sharded blockchains using product-form network queue (PFQN) and derived the maximum throughput of the sharded blockchain. Refs. [11,12] used an M/GB/1 queue model with batch service to analyze the transaction confirmation time in the Bitcoin system. Table 1 below is a summary of related work.

**Table 1.** Summary of Application Research of Queuing Theory in Blockchain Model Analysis.

| Reference | Methodology | Focus Area | Key Findings | Contributions to the Field |
|---|---|---|---|---|
| [8] | GI/M/1 queue with batch-service | Transaction confirmation in Single-chain systems | Developed a queueing theory model for blockchain systems, identifying average transaction numbers and confirmation times | Introduced an analytical approach for blockchain queueing systems |
| [9] | M/G/1 for delay characterization, machine learning for transaction classification | Transaction confirmation in Single-chain systems | Proposed a machine learning framework for transaction classification and queueing theory for delays | Enhanced understanding of blockchain delays and transaction confirmation dynamics |
| [10] | PFQN | Sharded blockchain efficiency | Established a model for sharded blockchain and derived maximum throughput | Introduced a new model for analyzing sharded blockchain performance |
| [11,12] | M/G/1 queue with batch service | Transaction confirmation time in Bitcoin | Analyzed transaction confirmation time in Bitcoin using queue theory | Applied queue theory to understand Bitcoin's transaction dynamics |

### 1.3. Motivation and Challenge

Brokerchain [13] found that, in Monoxide, 80,000 TXs are unevenly distributed across shards, with most TXs being cross-shard as the number of shards increases. This can cause infinite TX confirmation delays when the recipient account of a cross-shard TX is congested, which violates the principle of timeliness as defined in [14], where it is expected that a correct process will eventually write a valid transaction to its ledger. Another motivation stems from [10], who did not discuss the confirmation delay of a sharded blockchain. Moreover, because the cross-shard technique is introduced to sharded blockchains, meaning heterogeneity between shards and traditional blockchains, the theory in [11,12] is not applicable to sharded blockchains. This constitutes one of the motivations for our study, that these studies still lack a qualitative analysis of the confirmation time model for sharded blockchains. Addressing this issue by quantitatively characterizing the transaction-confirmation process is crucial for the scalability of sharded blockchains, a predominant direction in blockchain development. In this paper, we present the following contributions:

1.   We decouple the input of the sharded blockchain through the product-form queue network (PFQN) and solve the transactions at different stages to obtain the average expected value of the transaction confirmation time applicable to the sharded blockchain;
2.   We additionally consider the transaction confirmation process on the main chain, and add a new confirmation queue F after applying the PFQN model, making the model more in line with the actual transaction confirmation situation in the blockchain;
3.   We utilize the PFQN model to assess the impact of quantum-resistant technologies on sharded blockchain transaction times, enhancing security against quantum threats.

We provide a brief explanation of why we choose to use the PFQN model and give an overview of how the PFQN model operates in Section 2. Subsequently, in Section 3, we introduce the PFQN model in detail and extend it to derive the transaction-processing confirmation time in the system. Following that, we simulate the blockchain environment and analyze the impact of various parameters on the transaction confirmation time.

## 2. Materials and Methods

### 2.1. Why PFQN?

The PFQN model is particularly suited for analyzing sharded blockchain systems for several reasons, which relate directly to the characteristics and demands of sharded environments:

Product-form steady-state distribution: This characteristic means that the steady-state probabilities of the network can be factored into a product of simpler functions, each corresponding to a component of the network. In the context of sharded blockchains, this property is highly beneficial because it simplifies the analysis of complex systems. Sharded blockchains, by nature, are decentralized systems split into multiple shards (sub-networks), each processing its own set of transactions independently. The product-form characteristic allows for the analysis of each shard as an individual entity while still understanding its part in the greater system's dynamics.

Quasi-reversibility: Quasi-reversibility means that the queues within the network maintain a certain independence in terms of arrivals and departures. In sharded blockchain systems, this mirrors the operational independence of shards: each shard processes transactions independently but contributes to the overall system's throughput and latency. Quasi-reversibility makes it easier to predict overall system performance based on individual shard behaviors.

Scalability and decomposition: PFQN allows for the scalable analysis of networks, which aligns with the scalable nature of sharded blockchains. As blockchain systems grow and add more shards, the complexity increases. The PFQN model supports this scalability by enabling a modular approach to system analysis—each shard can be modeled separately but within the same framework, aiding in understanding the overall impact of scalability on the system performance.

Throughput and latency analysis: One of the key performance metrics for sharded blockchains is throughput (the number of transactions processed per time unit) and latency (the time taken for a transaction to be confirmed). The PFQN model is particularly adept at analyzing these metrics due to its focus on network queues and service processes. By applying the PFQN model to sharded blockchains, researchers can derive maximum throughput and expected latency, providing valuable insights into system efficiency and performance.

The PFQN model addresses the complexities of interacting shards more effectively than the GI/M/1 or M/G/1 models, which focus on simpler, single-chain systems. The PFQN model's effectiveness for sharded blockchains, shown in studies like [10], stems from its ability to model and analyze multiple shards, providing insights into throughput and inter-shard dynamics beyond single-queue analysis.

### 2.2. Blockchain Setting

In this work, we adopt settings similar to those used in [7], treating the Nakamoto consensus family as the intra-shard consensus mechanism, with cross-shard transactions employing the relay method.

In shard-based transaction methods like relay, the source shard verifies the input account's balance before the TX is relayed to the output shard. Relay checks cross-shard transaction accounts in blocks against transaction amounts. The cross-shard verification is termed the Availability Certificate (AC) (from Definition 32 in [15]). In our PFQN model, AC will also be referred to as a cross-queue signal in the following text.

### 2.3. Model Assumption

Our PFQN model is composed of a series of nonlinear QNs, with each QN queue consisting of a network queue and a consensus queue, as depicted in Figure 1. To ensure the model's accuracy and practicality, it is founded on a series of detailed assumptions that concern key aspects such as the arrival process and service mechanisms.

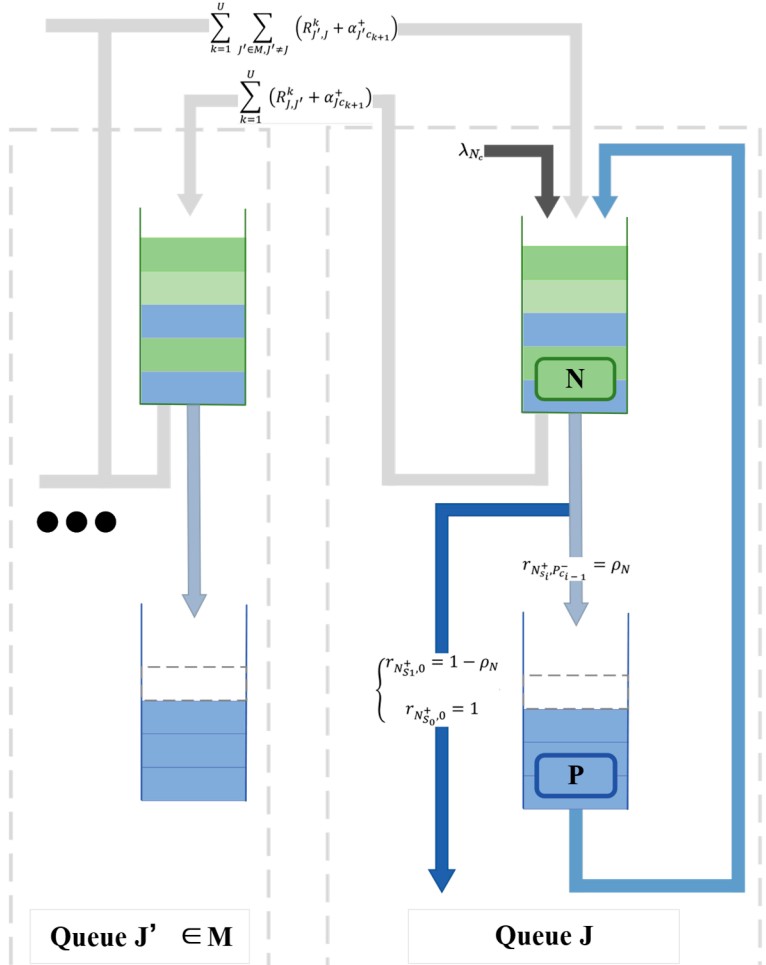

**Figure 1.** Queue network (QN) structure in PFQN.

In our model, we assume that the sizes of intra-shard transactions and ACs are independent of the number of their destination shard fields. This assumption might even be quite close to reality. In many instances, the bulk of a TX's size is occupied by the private signature of its sender, e.g., Bitcoin before the BIP1412 update [16]. It is reasonable to say that each transaction produces the same size for the shard.

We assume that the arrival of transactions to the network queue follows an independent Poisson process. In many existing projects, transactions are allocated to shards based on the sender address [7,17]. As a result, transactions generated by accounts are uniformly distributed across each shard. Given this, we assume that the rate at which transactions arrive at each shard is equal.

With transaction propagation and transaction arrival processes addressed, we can now begin to consider the processing capacity of the network queue. For network queues, due to their interaction with a shared medium in distributed systems, they are viewed as processor-sharing (PS) queues. This approach captures parallel information verification, thus leading to the classification of QNs as M/GB/1/PS queues. Similar assumptions have been adopted in previous work [10], where, under this assumption, by employing symmetric service rules (such as processor sharing in network queues), the quasi-reversibility (QR) property of queues is maintained even with non-exponential service time distributions. Processor sharing is a method of servicing multiple customers simultaneously by evenly distributing the service capacity to all current jobs. This principle helps maintain QR by ensuring that the service mechanism remains unbiased and symmetrical, allowing for the independence between arrivals, services, and departures required by QR, despite deviations from the exponential service time assumption.

After a transaction is processed by the network queue, each transaction routed by the network queue will leave the network queue and join the miner's mempool (consensus queue) after verification. Considering the exponential service distribution characteristic of PoW mining, as highlighted in previous studies [8,18,19], consensus queues are modeled as M/M/1/FCFS. This modeling approach takes into account the stochastic nature of mining and transaction processing within blockchain networks, where the service time for transactions (i.e., the time taken to mine a block and validate transactions) follows an exponential distribution.

The PFQN model's structure is set as open, where customers can leave the QN after receiving service and move to another QN, according to predefined routing rules. This structural assumption allows us to observe and analyze the dynamics of customer flow and the overall performance of the network.

### 2.4. Model and Derivation

### 2.4.1. PFQN Model

This discussion succinctly reviews how a transaction is confirmed in a sharded blockchain. A user-signed transaction is sent to a queue in a particular shard network, and the transaction is allocated to a specific shard based on certain rules (such as the hash value of the transaction). Once assigned to the corresponding shard, it enters the transaction pool maintained by the nodes of that shard, waiting to be selected for packaging into a block. Miners or validators in the shard select transactions from the pool and package them into a new block. This process occurs simultaneously across the network's various shards. Within each shard, a consensus mechanism is used to verify and confirm the new block. If a transaction involves cross-shard operations, it is first confirmed in the source shard. Subsequently, the transaction is relayed to other shards, and upon receiving the transaction information, the target shard verifies, executes, and confirms it.

In our nonlinear queueing networks, there are three distinct types of entities: regular customers, negative signals, and positive signals corresponding to customers.

There are five types of entity flows within the PFQN model, represented by $c$, $c_k^+$, $s$, $s_i^+$, and $c_i^-$.

The customer 's' represents a block component, and we refer to s here as a mini block, which contains only one transaction. A mini-block can represent a confirmed transaction and AC. We consider a mini-block instead of the block because mini-blocks can simplify the process of the coordinator extracting transactions from the block to generate a corresponding AC. The customer 'c' represents a transaction type customer, which in the context of blockchains, is a regular user-signed transaction.

To simulate batch service in the blockchain, we introduce $c_i^-$ and $s_i^+$. If a $c_i^-$ arrives at an empty queue, it will disappear. However, if a $c_i^-$ arrives at a queue with n customers, it will cause the customer at position l to leave with probability $\theta(l, n)$ such that $\sum_{l=1}^{l=n} \theta(l, n) = 1$. A higher-positioned c will fill the vacancy, triggering another $c_i^-$ at the

output of the queue. $s_i^+$ will trigger $s_{i-1}^+$ at the output of the queue while adding an s to the queue.

$c_k^+$ stands for cross-queue signal, and k in $c_k^+$ is the phase of the current signal. Stages are introduced to represent the number of shards yet to be visited by the signal. By replacing the concept of target sets in signals with stages, the probabilistic routing method models the process of cross-shard transaction transfer.

In Section 2, we have already made preliminary assumptions about the consensus queue P and the network queue N, which explain the distribution followed by the arrival and service processes of entities.

However, we still need to further explain the representation of arrival rates and the interactions between entities across queues. It is important to note that the arrival mechanism of entities in queue J is the same as that in its network queue N. Therefore, to simplify the discussion, we will no longer differentiate between the entity arrival processes in these two types of queues. In subsequent discussions, descriptions of entity arrivals may be used interchangeably, aiming to refer to this common arrival mechanism.

To facilitate the distinction between user-initiated transactions and relay's transaction arrival rate, we use the symbol $\lambda_{J_c}$ to represent the arrival rate of new customers in queue J. Here, $\lambda$ is a subset of $\alpha$, specifically denoting the rate at which new user-generated transactions arrive at queue J, i.e., $\lambda_J$. The arrival rates for queue J are represented by $\alpha_{J_{c_i}}^+$, $\alpha_{J_{s_i}}^+$, and $\alpha_{J_{c_i}}^-$, respectively.

After leaving a queue, each entity can change its type through network routing. For example, an entity u departing from queue J can become a v-type entity heading for queue J′ with probability $r_{J_u J'_v}$. The only requirement for routing probabilities is that $\sum_v \sum_u r_{J_u J'_v} = 1$.

Next, we will use two simple examples to explain how a regular customer (a user-signed transaction) and a cross-queue signal (a cross-shard transaction) are processed and transmitted within the PFQN. For the regular signal, we consider the propagation process of a signal within a single queue. For cross-queue signals, we will explore how a signal propagates through multiple queue systems, including the behavior of signals as they transfer between different queues. By describing the transfer process of signals in a single queue, we obtain an accurate description of the arrival rate of transactions to a queue in PFQN.

The way a regular customer operates in a queue can represent the confirmation process of a transaction within a shard. Customer c is first created by the client and propagated through the network to the shard's network queue N. Then, it enters N at rate $\lambda_{N_c}$. N distributes c to the nodes in the shard at a service rate $\mu_{N_c}$. Miners who have received c will add c to their transaction memory pool, representing c entering the shard's consensus queue P. The service rate $\mu_{N_c}$ represents the service rate of the transaction in the network. Since $\mu_{N_c}$ is large in reality, the service time can be negligible. Therefore, we simply see c entering queue P at rate $\lambda_{P_c}$.

When c reaches the end of P, as illustrated in Figure 1, the transaction first arrives at queue N and then reaches queue P at an extremely fast service rate. At this point, c is converted into signal $s_b^+$, represented by $r_{P_c, N_{s_b}^+}$. The signal then triggers a new s in N, transforming at the end of N to $c_i^-$. Here, i is equal to b-1 ($0 \leq i < b$), where b represents the size of a block, that is, the number of transactions a block can contain. When $c_i^-$ arrives at P, it will then cause the disappearance of the other c. Eventually, this process will remove b transactions from the node's mempool P, corresponding to a batch processing in the blockchain.

To satisfy quasi-reversibility, queues that receive positive signals must emit additional positive signals when empty. Therefore, we require network queues to emit positive signals whenever they do not contain block components. Following the approach in [10] to maintain the QR property, we adopt a probabilistic method to decide whether to retain the departing positive signals or route them out of the network. By multiplying by the reciprocal of a service rate, we adjust the emission rate of positive signals as the queue transitions between different occupancy states, especially when the queue is empty. This

adjustment compensates for the current load rate by emitting positive signals that maintain the QR property.

To ensure QR, $\alpha_{N_{s_i}^+}$ must be multiplied by $\rho_{Ns}^{-1}$ to adjust the rate of $\alpha_{P_{c_i}^-}$. However, to ensure that multiplying $\alpha_{N_{s_i}^+}$ by $\rho_{Ns}^{-1}$ does not deviate from the original scenario, we need to set $r_{N_{s_i}^+, P_{c_{i-1}}^-} = \rho_{Ns}$ and $r_{N_{s_1}^+, 0} = 1 - \rho_{Ns}$. In terms of service processes, $\mu_N$ represents the service rate of all entities in N. The utilization rate of queue N is represented as $\rho_{Ns} = \frac{\alpha_{Ns}}{\mu_N}$, which can be a combination of multiple category utilization rates. The total number of negative signals generated remains constant, so the queue is not affected by this setting.

We can derive the flow equations of the queueing network. Due to the symmetric architecture, we only need the equation of a shard, including the consensus queue and its related network queue. For $i = 1, \ldots, b - 1$, the flow equation of the consensus queue is:

$$\alpha_{Pc} = \rho_{Nc}\mu_{Nc} \tag{1}$$

$$\alpha_{P_{c_i}^-} = \rho_{Ns}^{-1}\alpha_{N_{s_{i+1}}^+}r_{N_{s_{i+1}}^+, P_{c_i}^-} = \alpha_{N_{s_{i+1}}^+} \tag{2}$$

The cross-queue signals mainly include the generation and transfer stages. When the positive signal $s_i^+$ arrives at N, the newly generated s is converted into a k-stage cross-queue signal $c_k^+$ at a certain rate, routing it to other queues besides itself. Once $c_k^+$ arrives and is processed, it continues to be routed as $c_{k-1}^+$ to other queues, excluding itself, until k equals 0. We consider shards j and j′ as examples, where $J′ \neq J$, $J′, J \in M$, $M = \{1, 2, \ldots, M\}$ is the set of all queues representing shards. For $k = 0, 1, \ldots, U$, and for all stages $k > U$, $\alpha_{J_{c_k}^+} = 0$, U is the maximum stage that the signal can reach. Given that the newly generated signal has the potential to impact a maximum of either M-1 or dmax (indicating the maximum destination that a signal can reach in one stage) shards, it follows that U = min(M-1, dmax)—1. We can obtain the overall arrival rate of cross-shard signals at shard J in stage k:

$$\alpha_{J_{c_k}}^+ = \lambda\delta[k] + \sum_{J' \in M, J' \neq J}\left(R_{J',J}^k + \rho_{J'c}^{-1}\alpha_{J'c_{k+1}}^+ r_{J'c_{k+1}^+, J_{c_k}^+}\right) \tag{3}$$

The three terms in $\alpha_{J_{c_k}}^+$ represent the arrival rate of $c_k^+$ in shard J, each term being one of the sources of $c_k^+$: the first term is the client-generated c arriving at shard J at rate $\lambda\delta[k]$ by P transforming into $c_k^+$. $\delta[.]$ is the Dirac function defined on the discrete domain. The second term is the signal $c_k^+$ generated by completing the block component service in the other shards J′. The third term is the signal $c_k^+$ routed from shard J′ to shard J with rate $\alpha_{J'c_{k+1}}^+ r_{J'c_{k+1}^+, J_{c_k}^+} \cdot \rho_{J'c}^{-1}$ is multiplied to prevent the additional departure rate.

The discussion will now focus more closely on the second and third items. During the generation stage, we need to consider the probability that a block component contains a cross-shard signal, as well as the probability of a cross-shard signal being at a certain stage. We need to differentiate between AC and TX in block component 's' to identify which components can be transformed into cross-queue signals. For this, we use $s_d, d = 1, \ldots, dmax$ to represent the ACs with d destinations. We define:

$$R_{J'J}^k = \rho_{J's}\mu_{J's}r_{J's, J_{c_k}^+} = \rho_{J's}\mu_{J's}\sum_{d=k+1}^{d_{max}}Pro(s = s_d)r_{J's, J_{c_d}^+} \tag{4}$$

as the rate at which the network queue of shard J′ generates block s at rate $\rho_{J's}\mu_{J's}$ and transforms it into $c_k^+$ to be routed to shard J at rate $r_{J's, J_{c_k}^+}$. The term $r_{J's, J_{cd}^+}$ represents the probability of routing to other shards and $\sum_{d=k+1}^{d_{max}}Pro(s = s_d)$ represents the probability that a block component contains a cross-queue signal of a certain stage. We know that all customers in a network queue are comprised of both "customers" that are newly issued by clients and "signals" routed from other shards. Hence, the probability that a block

component generates a signal can be derived as the ratio of the rate of newly issued TXs (i.e., $\lambda D[d]$) to the rate of all other customers in the network queue, i.e., $\sum_{k=0}^{U} \alpha_{Jc_k}^{+}$. Thus:

$$\Pr(s = s_d) = \frac{\lambda D[d]}{\sum_{k=0}^{U} \alpha_{Jc_k}^{+}}$$

To obtain the routing probabilities $r_{J's, J_{c_d}^{+}}$, the first step is to find the number of distinct shards other than the source shard that a multi-destination TX points to. The number of sets with i ($i \leq d$) distinct shards other than the originating shard in the destination fields of $s_d$ is

$$N(|M|, d, i) = \frac{(|M|-1)!}{(|M|-i-1)!} \begin{Bmatrix} d+1 \\ i+1 \end{Bmatrix}$$

where

$$\begin{Bmatrix} d+1 \\ i+1 \end{Bmatrix} = \frac{1}{(i+1)!} \sum_{p=0}^{i+1} (-1)^p \binom{i+1}{p} (i+1-p)^{d+1}$$

is the second kind of Stirling number, which is the number of ways to partition a set of d+1 objects into i+1 non-empty subsets. Therefore, routing probability $r_{J's, J_{c_d}^{+}}$ is obtained by dividing $N(M, d, i)$ by $M^d$ possible destination sets for $s_d$.

Owing to the population dynamics within the target shards, aside from the source shard, where the newly emerged signal may be directed, this necessitates the division of $\rho_{Ns}\mu_{Ns}$ by $|M|-1$. Given the symmetrical and identical nature of the queues within M, $\rho_{J's}\mu_{J's} = \rho_{Js}\mu_{Js}$, it is true that $\rho_{J's}\mu_{J's}$ can be simplistically represented as $\rho_{Js}\mu_{Js}$. By incorporating these equations into Equation (4), we derive:

$$R_N^k = \frac{\rho_{Ns}\mu_{Ns}}{|M|-1} \frac{\lambda}{\sum_{k=0}^{U} \alpha_{Nc_k}^{+}} \sum_{d=k+1}^{d_{max}} \begin{Bmatrix} d+1 \\ k+2 \end{Bmatrix} D[d] \frac{\prod_{z=1}^{k+1}(|M|-z)}{|M|^d}, k = 0, 1, ..., U \quad (5)$$

During the transfer stage, consider $c_i^{+}$ and $\alpha_{Jc_i}^{+}$ as the multi-stage positive signals and their respective arrival rates, where i represents the stage. When a $c_i^{+}$ enters the network queue, it not only adds a class c customer to the queue but also the newly triggered signal is routed as $c_{i-1}^{+}$. If the stage of the signal is 1, then the signal is routed as a regular class c customer. Due to uniformly distributed routing probabilities, it can be routed to any of the other M − 1 shards with equal probability.

$$\rho_{J'c}^{-1} \alpha_{J'c_{k+1}}^{+} r_{J'c_{k+1}^{+}, c_k^{+}} = \rho_{J'c}^{-1} \alpha_{J'c_{k+1}}^{+} \left( \frac{\rho_{J'c}}{M-1} \right) = \frac{\alpha_{J'c_{k+1}}^{+}}{M-1}, \quad (6)$$

Due to the symmetric structure and flow of each shard, each shard equally hosts the same rate of multi-destination TXs as the others. Hence, both rates in the summation of Equation (3) are independent of their originating queues. Therefore, we can simply replace the subscript J′ with J in $\alpha_{Jc_{k+1}}^{+}$ and rewrite it as $\alpha_{Nc_{k+1}}^{+}$, then, we replace Equation (3) with Equation (5) and obtain:

$$\alpha_{Nc_k}^{+} = \lambda \delta[k] + (M-1)R_N^k + \alpha_{Nc_{k+1}}^{+} \quad (7)$$

where $R_{N_k}$ is the rate at which the transactions are processed. Starting to solve (7) from k = U down to k = 0, we can obtain the total input rate of combined-flow customers to a network queue as $\lambda_{all}$

$$\lambda_{all} = \Sigma_{k=0}^{U} \alpha_{Nc_k}^{+} = \frac{\frac{\rho_P(1-\rho_P^b)}{1-\rho_P} *}{\mu_P} \frac{}{1+\sum\limits_{k=0}^{U} (k+1)\sum_{d=k+1}^{d_{max}} \begin{Bmatrix} d+1 \\ k+2 \end{Bmatrix} D[d] \frac{\Pi_{z=1}^{k+1}(M-z)}{M^d}} \tag{8}$$

### 2.4.2. Derivation of Transaction Confirmation Delay

Using the PFQN model, we decouple the input model of the sharded blockchain, and we sum entities c in different stages to obtain the average expected value of transactions applicable to the sharded blockchain. However, obtaining a description of a queue's transaction flow is not sufficient to determine the transaction confirmation time for a queue. By utilizing the formula described in [12] for the confirmation time of transactions in a single queue and combining it with the decoupled transaction entities, we have derived the expected confirmation time required for a cross-shard transaction.

We defined the block generation time E(S) as the time interval between consecutive block-confirmation time points. We also regard a block generation time as a service time. Let Si denote the ith block generation time. Similar to numerous studies [11,12,20,21], we consider the block generation time of PoW to follow an exponential distribution. Therefore, we define the block generation time S as adhering to the exponential distribution, described by the following formulation:

$$G(x) = 1 - e^{fx},$$

It is assumed that the sequence {Si} consists of independent and identically distributed (i.i.d.) random variables, each characterized by the distribution function G(x). Let g(x) denote the probability density function of G(x). The mean block generation time E[S] is given by

$$E[S] = \int_0^\infty xg(x)dx.$$

$$E[S] = \frac{1}{f}, \quad E[S^2] = \frac{2}{f^2}$$

Let $\zeta(x)$ denote the hazard rate of S, which is given by $\zeta(x) = \frac{g(x)}{1-G(x)}$.

T denotes the transaction confirmation time, i.e., the time interval between when the user issues a transaction and when a block containing the transaction is generated ([15] Definition 26). Let Num(t) denote the number of entities c in P at time t, and X(t) denote the elapsed service time at t. We attain $P_n(x,t) = \frac{d}{dx}Pr\{Num(t) = n, X(t) \le x\}, P_n(x) = \lim\limits_{t\to\infty} P_n(x,t)$. Given Little's theorem, we know that the long-term average number of customers (E(N), or the expected transaction volume) is equal to the long-term effective arrival rate ($\lambda$, or the speed at which transactions arrive at the system) times the average waiting time of customers in the system (E(T), or the transaction confirmation delay). The average transaction confirmation time can be given by $E[T] = \frac{E[N]}{\lambda}$.

We next introduce the entity concept into an important formula ([12] Theorem 1) to find the entity confirmation time.

$$E[T]_e = \frac{1}{2\lambda^2(b-\lambda E[S])} * \tag{9}$$

$$\left(\sum_{k=1}^{b}\beta_k\left[b(b-1)+\{(b+1)b-k(k-1)\}\lambda E[S]+(b-k)\lambda^2 E\left[S^2\right]\right]\right.$$
$$\left.-\lambda\left\{b(b-1)-\lambda^2 E\left[S^2\right]\right\}\right)$$
$$\beta_k = \int_0^\infty P_k(x)\zeta(x)dx$$

$\beta_k$ represents the probability that P has k-1 entities during the entire system runtime. This reveals the entities' confirmation time when $\Sigma_{k=0}^{U}\alpha_{Nc_k}^{+}E[S] \leq b$, meaning that the system is stable. In a system comprising M queues, each conforming to a quasi-reversible M/M/1 queue model, the composite arrival process at an individual queue retains the characteristics of a Poisson process. This holds under the condition that each customer, upon service completion, has a probability r of being routed to any other queue in the system, with each of these queues having an equal probability of $\frac{1}{N-1}$ of receiving the customer. Recall that an entity with k stage arrives at P according to a uniform Poisson process with rate $\alpha_{Nc_k}^{+}$ across all queues. Therefore, we apply this theorem to a synthetic flow queue P, with $\Sigma_{k=0}^{U}\alpha_{Nc_k}^{+}$ satisfying Poisson distribution.

However, applying (7) directly to (9) will only give the expected time $E[T]_e$ for a $c_k^{+}$ to be processed. Recall that our goal is to get the expected time for a TX, so this does not meet our expectations. Knowing that $E[T]_e$ is the average expected time for c to complete the service in queue P or the average expected time for $c_k^{+}$ to accept the service and transform into $c_{k-1}^{+}$, we can obtain the expected service time for a transaction to accept service in the QN queue:

$$E[T]_{process} = \Sigma_{k=0}^{U}\alpha_{Nc_k}^{+}kE[T]_e \tag{10}$$

(10) reveals that new arrivals are multiplied by their numbers in the target fields. Since it needs to be executed sequentially k times in different shards, according to the definition of an eventual sharded blockchain in ([15] Definition 29), a transaction or block does not confirm instantly, and several blocks at the end of a blockchain must be added to obtain stable states. Therefore, $E[T]_{process}$ cannot represent the expected delay in transaction confirmation, because the PFQN model was designed according to PoW consensus within the shard and the cross-shard consensus relay method, so it should meet the definition of an eventual sharded blockchain. Although the PFQN model is very applicable to an eventual sharded blockchain, the model still needs to introduce a new queue to simulate the confirmation time of transactions in the shard's main chain.

We additionally considered the stabilizing process of transactions on the main chain by adding a new confirmation queue F at the end of queue N which is more consistent with the actual situation of transactions being confirmed on the blockchain. F is an M/M/1 queue, i.e., both arrival and service processes follow a Poisson distribution, as shown in Figure 2. The arrival rate $\lambda_F$ of queue F includes two entities, $s_0^{+}$ and $s_1^{+}$ from queue N. It is obvious that $\lambda_{F_s} = \mu_{N_{s1}^{+}}$. The confirmation queue F processes block component s with a service rate $\mu_F$. The average processing and waiting time of the block component, which is also the confirmation time of the transaction on the main chain, can be obtained through the waiting time formula $T_F = \frac{1}{(\mu_F - \lambda_F)}$. Substituting $\lambda_F = \mu_{N_{s1}^{+}} = \alpha_{Jc_1}^{+} = \lambda + \Sigma_{k=1}^{U}(M-1)R_J^k$, we get

$$T_F = \frac{1}{\mu_F - \lambda - \Sigma_{k=1}^{U}(M-1)R_J^k} \tag{11}$$

Using (10) and (11), the time from a transaction being issued to being fully confirmed, E[T], can be calculated as

$$E[T] = E[T]_{process} + T_F = \Sigma_{k=0}^{U}\alpha_{Nc_k}^{+}kE[T]_e + \frac{1}{\mu_F - \lambda - \Sigma_{k=1}^{U}(M-1)R_J^k}$$

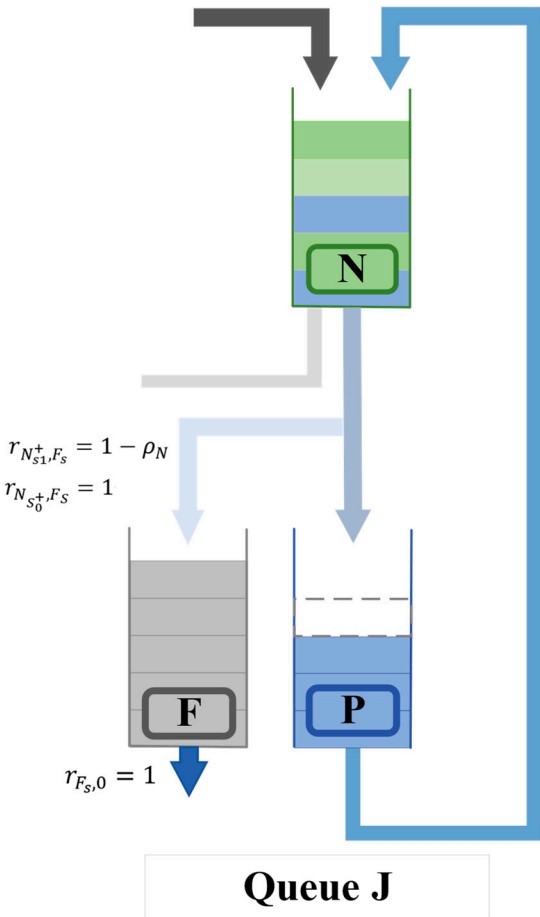

**Figure 2.** Queue network structure with confirmation queue.

### 3. Results

We simulate the sharded blockchain as a PFQN queue. When pairs are in the stationary state, we analyze the impact of different variables on shard $\lambda$. To analyze $\lambda$, we need to first obtain the initial values of different parameters, D(d), as the distribution law of the transaction set $\text{Txs}_d$. We give the following definition:

$$D(d) = [\Phi_1, \ldots, \Phi_d, \ldots, \Phi_{dmax}], 1 < d < dmax$$
$$\Phi_d = \frac{\text{Txs}_d}{\text{Txs}}$$

Because of the need to carry out $\sum_{k=1}^{d} k\Phi_k$ piecewise selection, it is equivalent to carrying out $\sum_{k=1}^{d} k\Phi_k$ i.i.d. random experiments, and the number of times each piecewise is selected is subject to binomial distribution $\text{Bin}\left(\sum_{k=1}^{d} k\Phi_k, \frac{1}{M}\right)$. According to the central limit theorem, the binomial distribution can be approximated by a normal distribution when the number of trials is large enough. In the Bitcoin and Ethereum marketplaces, we know that the number of transactions is large enough, so we assume that D(d) is normally distributed. The expectation of and variance in the binomial distribution gives us D(d) obeying N $\left(\frac{\text{TX}_{\text{NUM}}}{M}, \sqrt{\left(\frac{\text{TX}_{\text{NUM}}(M-1)}{M^2}\right)}\right)$. Here, we assume that dmax is a constant, and in practical UTXO scenarios, each transaction usually involves a finite number of inputs and outputs. For example, a standard Bitcoin transaction typically contains 2.26 UTXOs with a small difference, possibly around 1. Ref. [22] draws this conclusion of basic facts of the analyzed UTXO set. We set the number of shards as five, the utilization rate $\rho p$ as 0.995, block b as containing five transactions each time, and the maximum degree of the transaction dmax as two.

The following is the simulation of the sharded blockchain under the theory of the PFQN confirmation model.

Figure 3 shows the effects of utilization rate (ρp) and transaction degree (dmax) on arrival rate (λ). Figure 3a reveals an exponential increase in λ with ρp, highlighting capacity near-saturation effects. Figure 3b depicts a decline in λ with increased dmax, stabilizing beyond a certain complexity level, indicating an initial efficiency drop that plateaus.

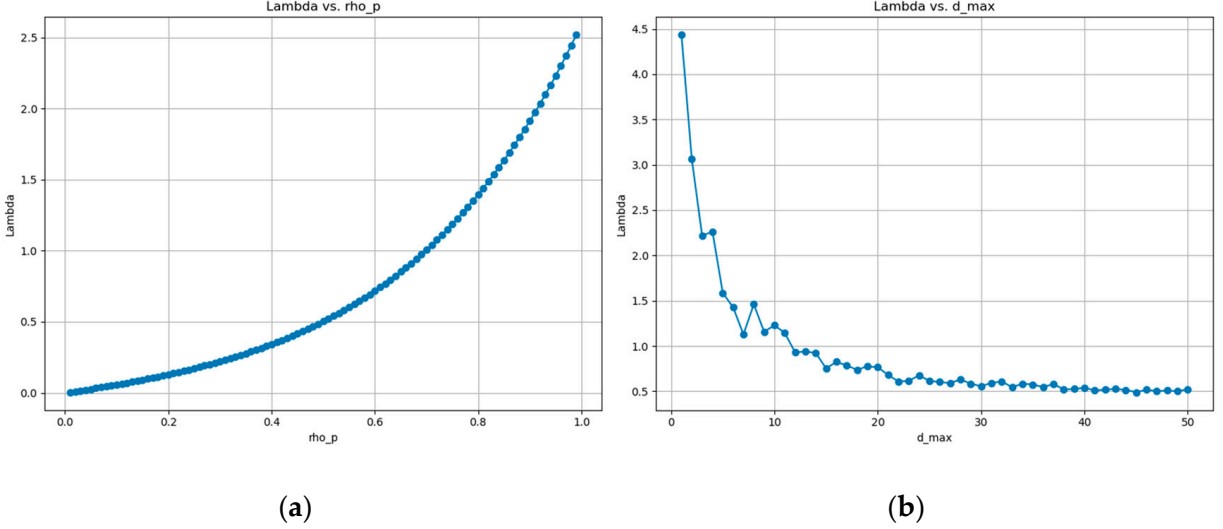

(**a**)　　　　　　　　　　　　　　　　　　　　(**b**)

**Figure 3.** (**a**) Impact of ρp on arrival rate λ; (**b**) impact of transaction degree dmax on arrival rate λ.

Figure 4 examines the impacts of shard count (M) and block size (b) on λ. Figure 4a illustrates a significant decrease in λ with higher M, plateauing after reaching a certain number of shards, suggesting initial efficiency gains that level off. Figure 4b demonstrates a consistent increase in λ with larger b, indicating linear scalability with block size.

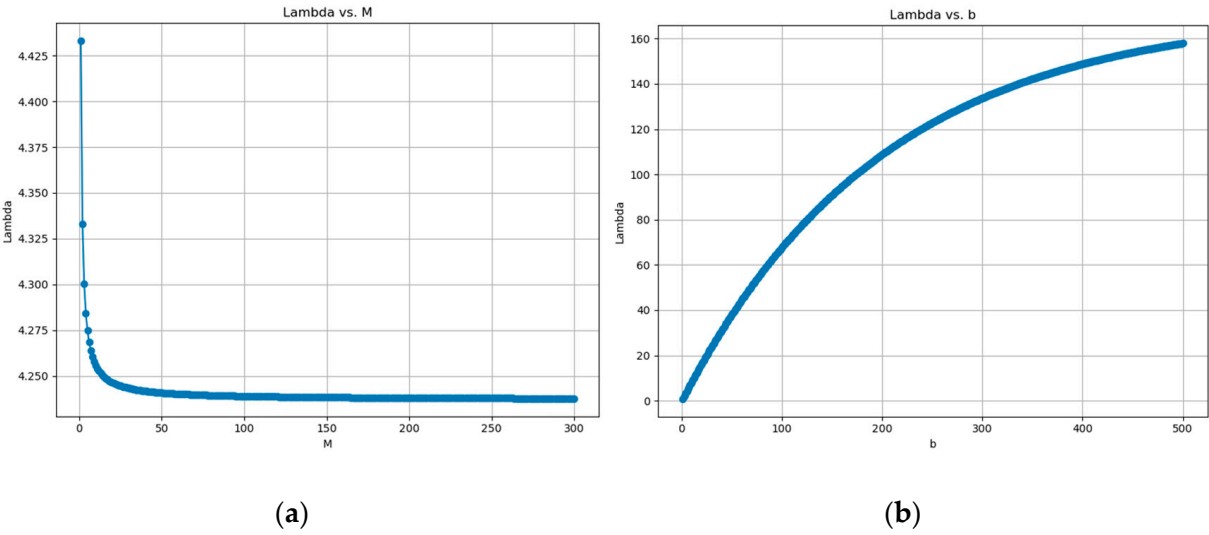

(**a**)　　　　　　　　　　　　　　　　　　　　(**b**)

**Figure 4.** (**a**) impact of number of shards M on arrival rate λ; (**b**) impact of block size b on arrival rate λ.

Figure 5 presents a surface plot of λ and expected service time E(S) against expected confirmation time E(T), showing a steep increase in E(T) with higher λ, especially at low E(S). This illustrates the critical balance between transaction processing capabilities and load management for maintaining reasonable confirmation times in sharded blockchains.

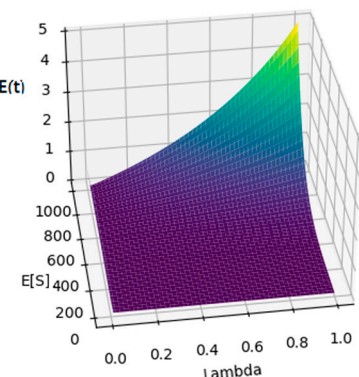

**Figure 5.** Effect of λ and E(S) on E(T).

In our simulate experiment, we utilize BlockEmulator [23] to simulate the real transaction confirmation latency in sharded blockchain systems. This decision is based on BlockEmulator's sophisticated ability to replicate the complex operations and network conditions of sharded blockchains accurately. Its detailed emulation of transaction processing, consensus mechanisms, and inter-shard communication provides a realistic environment in which to measure and analyze TX confirmation latency. Additionally, its capability to mimic real-world network scenarios, including packet transmission and bandwidth constraints, ensures that our latency measurements reflect practical blockchain conditions. This makes BlockEmulator an essential tool for our research, offering valuable insights into optimizing the transaction efficiency and scalability in sharded blockchain architectures.

In our study, we ensure that each shard queue remains stable, meaning that, for $\lambda_{all}$, the expected service time E[S] is less than the arrival rate λ. Figure 6 illustrates the transaction latency in a simulated sharded blockchain environment under real transaction conditions, with 100,000 Ethereum transactions injected at a constant rate. The simulations were performed with different numbers of shards, specifically 2, 4, 50, and 100, while maintaining the number of nodes within each shard at four. The figure compares the transaction delays within the sharded blockchain with the expected delays across different numbers of shards.

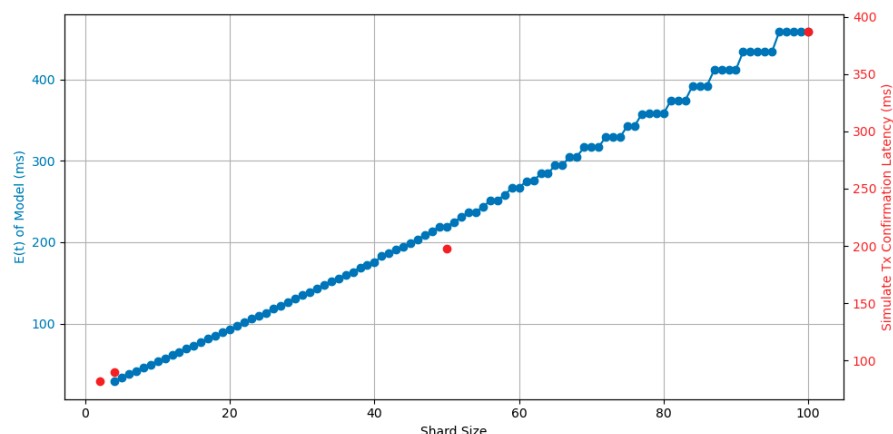

**Figure 6.** Theoretical model of PFQN and simulated data of block emulator.

Figure 6 illustrates how E(T) varies with shard size, with both simulation results and a theoretical model displayed. As shard size increases, E(T) follows a rising trend, indicating longer confirmation times with larger shards.

## 4. Discussion

### 4.1. PFQN and Sharded Blockchain Simulation

In the exploration of sharded blockchain systems, our study identifies critical parameters influencing the system throughput λ and overall performance. Notably, Figure 3a elucidates the positive relationship between the participation rate ρp and system throughput λ, signifying that enhanced participation in the blockchain network correlates with increased throughput.

Conversely, Figure 3b presents a contrasting scenario where an increase in the number of shards involved in a transaction inversely affects the system throughput. This decline is attributable to the augmented coordination costs inherent in managing multiple shards.

Further complications arise as delineated in Figure 4a, where augmenting the number of resources or shards correspondingly diminishes the λ attainable by a single shard. This decrement underscores the dilutive effect of resource distribution across an expanded set of shards, implying the importance of resource allocation efficiency. Conversely, Figure 4b illustrates a logarithmic increase in the system throughput λ as the transaction size (b) processed per consensus round is amplified. This suggests that, while larger transactions impose more significant processing demands, their integration into consensus rounds significantly boosts throughput.

Our investigation extends to the system performance metrics, E(T) and E(S), as depicted in Figure 5. An increase in the system performance index λ exhibits a concomitant rise in E(T), indicating a positive correlation between the system throughput and the expected time for transaction processing or consensus attainment. This positive association may stem from the enhanced complexities or delays engendered by elevated transaction rates or consensus challenges as throughput escalates.

Similarly, the correlation between E(S) and transaction confirmation time illuminates the impact of block production timelines on the transaction latency. An elongation in block generation duration necessitates that transactions endure extended confirmation periods, awaiting the endorsement of succeeding blocks. Therefore, optimizing both the λ and E(S) emerges as paramount in facilitating rapid transaction confirmation.

Nevertheless, Figure 6 unveils discrepancies potentially ascribable to the transaction allocation process within the sharded architecture. The deviations observed could emanate from the challenges inherent in replicating an idealized uniform arrival flow and constant service rate within a dynamic, real-world environment. In summation, our findings advocate for a balanced approach to sharded blockchain design, where the imperative to optimize the throughput and security is counterbalanced by the necessities of efficient resource utilization and strategic transaction-size management.

### 4.2. Security Analysis

In our research, we focus on the transaction latency of sharded blockchains, in particular simulating the transaction confirmation process through the PFQN model. The PFQN model, as a tool for analyzing the transaction confirmation process, could theoretically be used to evaluate scenarios that contain quantum resistance mechanisms. Assuming that quantum-resistant digital signature and encryption algorithms are implemented in a sharded blockchain, we can use the PFQN model to simulate and quantify the potential impact of these quantum-resistant measures on transaction confirmation times.

#### 4.2.1. Prior Research on Quantum-Safe Blockchain

With the advancement of quantum computing, there are increasing challenges to the security of blockchain technology, particularly the vulnerability of traditional blockchains to quantum algorithms. Consequently, we have integrated various research findings into model modules to study the transaction confirmation time of the PFQN model in the context of quantum computing.

Existing studies [24] have ensured security through three main aspects: data, transmission, and verification. Specifically, qBitcoin utilizes quantum transmission technology,

employing quantum teleportation for currency transmission. This ensures that once the currency is sent, the sender cannot retain the original currency data, effectively preventing double-spending issues. Furthermore, qBitcoin employs quantum digital signatures to verify transactions, requiring other participants to validate the signatures, thus maintaining compatibility with the principles of peer-to-peer (P2P) cash systems. In terms of data transmission, qBitcoin uses the Quantum Key Distribution (QKD) protocol to share private keys with the receiver.

Regarding data transmission and verification, ref. [25] utilized quantum one-way functions based on the Quantum State Computational Distinguishability (QSCD) problem to design quantum asymmetric encryption algorithms, ensuring the security of the verification process. This method effectively prevents eavesdropping, forgery, denial, and interception attacks. Additionally, witness nodes selected through the DPoSB (Delegated Proof of Stake based on node behavior and Borda count) mechanism are responsible for verifying transaction signatures. Ref. [26] also analyzed two lattice-based post-quantum encryption schemes.

### 4.2.2. Attack Models and Assumptions

Assuming that an adversary possesses a super quantum computer with over 1000 error-corrected qubits and low decoherence times, it could feasibly compute $10^{12}$ true random numbers per second using Grover and Shor algorithms, surpassing the current classical methods.

In the context of quantum computers, the security of blockchains is under a double threat [27,28]. On the one hand, the acceleration of Grover's algorithm [29] regarding the search problem may cause some operations in the blockchain network to occur faster than expected. On the other hand, Shor's algorithm [30] threatens potential damage to the traditional encryption method, which may lead to the security of the private key no longer being guaranteed. We demonstrate security in the malicious attacker model in Table 2, according to the Shor and Grover algorithms.

**Table 2.** Attack Scheme.

| Attack Type | Affected Blockchain Component | Attack Purpose | Means of Attack |
|---|---|---|---|
| Block Replacement Attack | Blockchain Historical Records | To replace the existing blockchain rewrite historical records. | Using Grover's algorithm to calculate nonces |
| Signature Forgery Attack | Transaction and Message Signatures | To tamper with or forge transactions | Using Shor's algorithm to break public key encryption systems |

First, we discuss the utilization of the Grover algorithm for executing a single block replacement attack within Bitcoin. As mentioned, for Bitcoin, the Grover algorithm reduces the number of attempts required to find a valid block from $2^{64}$ to about $2^{32}$.

In such an attack scenario, we assume there is a quantum computer capable of executing one trillion ($10^{12}$) attempts per second. Theoretically, this machine could complete $2^{32}$ attempts per second. Therefore, under ideal conditions, it could find a Bitcoin block in 0.0043 s. If this quantum computer is used to execute a block replacement attack, it could replace six blocks in 0.0258 s. Once the length of the attacker's private chain exceeds the official chain, the network's nodes will accept this private chain according to the principle of the longest chain, resulting in the original blockchain being overwritten. This would allow attackers to rewrite transaction history, potentially leading to double-spending attacks.

For an ongoing transaction 'c', if it is included in the block replaced by the attackers, this transaction might disappear from the blockchain because the attackers may not include it in their reconstructed blockchain. This means a transaction might never be confirmed due to the attack. Meanwhile, since the attacked shard transaction becomes invalid, all transactions involving this cross-shard might fail to be completed.

Existing strategies to counter this attack involve adjusting the difficulty level to make it hard for quantum computers to compute, introducing problems due to com-

putational power imbalance. Another strategy is to adopt a reputation model, introducing a penalty mechanism which promptly replaces block producers when a block replacement attack occurs, punishing malicious nodes. Regardless, the solutions include improvements to the consensus mechanism and adjustments to the difficulty, modifying the block generation mechanism.

For the signature forgery attack, we focus on the verification of the protocol, rather than the whole protocol covering transmission, data processing, and verification. The reason for this is that the transport and data processing steps are heavily dependent on the specific protocol code and data format, and their complexity is beyond the scope of this article. On the contrary, the verification link covers the integrated application of cryptographic algorithms and is the core of blockchain security under a signature forgery attack. In this study, the encryption algorithms adopted in the verification phase will be explored in detail, and in particular, their computational complexity against signature forgery attacks will be evaluated as a basis for measuring their security metrics. With this focus, we provide a methodology for assessing the overall security of a system without delving into the specific details of the protocol.

We discuss, in this section, the following two post-quantum encryption algorithms integrated into the PFQN model.

We first measure the security performance of the blockchain by the computational complexity of the encryption algorithm and reflect it in the expected transaction time. We use the National Institute of Standards and Technology (NIST) security levels to measure how hard an encryption algorithm is to break. The NIST is rigorously working to analyze, test, and validate post-quantum algorithms and is expected to release a draft standard by 2023. We can see the encryption difficulty corresponding to different NIST levels in Table 3.

**Table 3.** NIST level.

| NIST Level | Encryption Standard |
|---|---|
| 1 | AES 128 |
| 2 | SHA3-256 |
| 3 | AES192 |
| 4 | SHA3-384 |
| 5 | AES256 |

We then refer to the NIST level of post-quantum encryption algorithms in [31,32] as shown in Table 4.

**Table 4.** Cryptographic algorithm and corresponding difficulty.

| Algorithm Category | Cryptographic Algorithm | Private Key Length (bytes) | Public Key Length (bytes) | NIST Level | Approximate Probability of Compromise |
|---|---|---|---|---|---|
| Post-quantum encryption algorithm | CYSTAL-Dilithium3 | 1952 | 4000 | 3 | $2^{-192}$ |
| | FALCON | 1793 | 2305 | 5 | $2^{-256}$ |
| Classic | RSA | 3072 | 3072 | 1 | $2^{-128}$ |
| | ECDSA | 256 | 512 | 1 | $2^{-128}$ |

Theorem A2 (proof provided in the Appendix A) is proposed for the analysis of the security lower bound. In the theorem, the parameter h relates to the safety parameter of the encryption algorithm, representing the probability that the encryption algorithm can be successfully attacked through signature forgery. $|P_i|$ represents the total length of transactions processed by the ith shard, with the expected confirmation time denoted as E[T], recalling that U denotes the maximum number of shards a single transaction can involve.

Then, we map the security coefficient h to the computational complexity of the encryption algorithm and analyze the security of the PFQN model under the same configuration. We define security as the expected number of rounds a shard blockchain system can

safely process transactions before the first occurrence of a security vulnerability or unsafe transaction caused by quantum computational capabilities, in the face of attacks based on Shor's algorithm.

In the context of quantum computing, to ensure the PFQN model maintains the same security standards as Bitcoin, $\varepsilon$, representing the likelihood of a transaction's successful execution with cryptographic protection, is set to $2^{-128}$. Furthermore, h is defined as the requisite number of computations a quantum computer must perform on the encryption algorithm. This adjustment aligns with the security parameters necessary for safeguarding transactions within sharded blockchain systems against the computational capabilities of quantum computing. Then, we can use Theorem A2 to calculate the expected upper bound of the total number of secure transactions for the PFQN model integrated with the encryption algorithm under the NIST framework for shard blockchains. Finally, assuming that the service rate $\mu_p$ of the consensus queue P is linearly related to the performance of the encryption algorithm [32] (see Table 5), from Formula (8), it is known that $\lambda_{all}$ is also linearly related to $\mu_p$. We can calculate the expected total time for all transactions in the shard blockchain under the guarantee of security.

**Table 5.** Security bounds and performance of cryptographic algorithms.

|  | Signatures/s | Verifications/s | Max Safe Transactions | Expected Encryption Time per Transaction |
|---|---|---|---|---|
| CYSTAL-Dilithium3 | 6506.33 | 17,561.33 | $\approx \frac{2^{21}}{U}$ | 0.000154 |
| FALCON | 1446.52 | 9782.67 | $\approx \frac{2^{85}}{U}$ | 0.000691 |

In the context of quantum attacks, these expected upper bounds signify the maximum number of transactions that can theoretically be executed safely. Taking into account the time for each transaction, we can infer that, within these security limits, a system using the CRYSTAL-Dilithium3 algorithm could process a large number of transactions and blocks very rapidly. However, due to its lower upper limit, if it is to be used in a shard blockchain, stronger security parameters must be employed to enhance the algorithm's resistance to quantum attacks, thereby increasing the upper limit of secure transactions. This approach may sacrifice some performance, as stronger security parameters typically result in larger signature sizes and longer processing times. Conversely, the FALCON algorithm has a longer processing time for individual transactions, and its optimization goals should focus on improving algorithm performance. Recall that, from Figure 5, as $\mu$ decreases, E(S) causes E(T) to increase exponentially.

## 5. Conclusions

In this paper, we introduce the PFQN model to solve confirmation latency in sharded blockchains. Our analysis highlights the interplay between the network load, consensus efficiency, and security in sharded blockchains, providing insights for enhancing their robustness and efficiency. This work not only addresses scalability but also paves the way for future research, with plans to test our model in various scenarios.

**Author Contributions:** J.W.: Conceptualization, Methodology, Prototype Development, and Writing—Original Draft Preparation. W.R.: Methodology, Writing—Review and Editing, and Supervision. H.D.: Methodology, Writing—Review and Editing, and Supervision. J.C.: Writing—Review and Editing. All authors have read and agreed to the published version of the manuscript.

**Funding:** This paper is funded in part by the National Natural Science Foundation of China (62032019, 61732019, 61672435), and the Capacity Development Grant of Southwest University (SWU116007).

**Data Availability Statement:** Most data is contained within the article. All the data are available on request due to restrictions, e.g., privacy or ethics.

**Acknowledgments:** The authors are grateful to the RISE Laboratory at Southwest University for their support and contributions to this work.

**Conflicts of Interest:** The authors declare no conflicts of interest.

## Nomenclature

| Parameter | Description |
|---|---|
| M | Set of queues |
| $\lambda$ | Customer input rate per shard |
| d | Total number of destination fields in a TX |
| D[d] | The probability distribution for 'd' |
| b | Maximum number of TXs allowed in a block |
| c | Regular customer |
| $c_k^+$ | K stages cross-shard signal |
| s | Block components |
| $\alpha_{Je}$ | The arrival rate of customer type e to queue J |
| $\alpha_{Js_i}^+$ | The arrival rate of positive signal $s_i^+$ to queue J |
| $\alpha_{Jc_i}^+$ | The arrival rate of cross-shard positive signal $c_i^+$ to queue J |
| U | Maximum stage achievable for signal $c_i^+$ |
| $R_{J'J}^k$ | The service completion rate for a receipt in network queue J′ leading to a stage k signal $c_k^+$ for network queue J |
| $\mu_{Je}$ | Service rate for customer type e in a standard queue J |
| $\rho_{Je}$ | The utilization factor incurred by customer type e on a typical queue J |

## Appendix A

Before we begin our proof, we need to introduce the definition of blockchain security by referring to previous research [15,33]. In order to prevent readers from confusing the related concepts in the PFQN model, it is necessary here to prove security with a new set of symbols.

Beginning with the identification of key parameters in the security definition: $\mu$ represents the ratio of honest blocks in the shard chain, and k is identified as the safety coefficient in the state machine replication protocol.

**Definition A1 (A Secure Sharding Blockchain).** *Let $(A, Z)$ be an adversary and environment pair w.r.t. a sharding consensus protocol $\Pi$. $T_{initial}$ denotes the time for a sharding blockchain protocol to start up, including the production of genesis blocks and initial committees. $T_{liveness}$ denotes the transaction confirmation delay parameter, i.e., the time required to commit a transaction. We say $\Pi$ is secure w.r.t. $(A, Z)$ with parameters $T_{initial}$, $T_{liveness}$ if the following properties hold with an overwhelming probability:*

**Definition A2 (Consistency).** *Consistency includes the following two properties*:

Common prefix inside a shard: For any two honest nodes $i, j \in \text{shard}_S$ where $S \in [1, M]$, node i outputs $\text{LOG}_i$ to Z at time t, and node j outputs $\text{LOG}_j$ to Z at time $t'$, it holds that either $\text{LOG}_i \leq \text{LOG}_j$ or $\text{LOG}_j \leq \text{LOG}_i$.

No conflict between shards: For any two honest nodes $i \in \text{shard}_s, j \in \text{shard}_{s'}$ where $s, s' \in [1, m]$ and $s \neq s'$, node i outputs $\text{LOG}_i$ to Z at time t, and node j outputs $\text{LOG}_j$ to Z at time $t'$. For any transaction $tx_1 \in \text{LOG}_i$ and $tx_2 \in \text{LOG}_j$ where $tx_1 \neq tx_2$, it holds that $tx_1$ and $tx_2$ do not conflict with each other, i.e., there is no input that belongs to $tx_1$ and $tx_2$ simultaneously.

$$\neg\big((tx_1 \in \text{LOG}_i \wedge tx_2 \notin \text{LOG}_j) \vee (tx_1 \notin \text{LOG}_i \wedge tx_2 \in \text{LOG}_j)\big)$$

**Definition A3 (Liveness).** *For any honest node from any shard, if it receives a transaction TX at* $\text{time} t_0 \geq T_{initial}$ *from* $\mathcal{Z}$, *then at* $\text{time} t_0 + T_{liveness}$, *TX must be accepted or rejected.*

**Definition A4 (Persistence).** *Parameterized by* $k \in \mathbb{N}$ (*"depth" parameter*), *if in a certain round an honest party reports a shard that contains a transaction TX in a block at least k blocks away from the end of the shard's ledger (such transaction will be called "stable"), then whenever TX is reported by any honest party it will be in the same position in the shard's ledger.*

**Assumption A1.** *In the following discussion, the consensus protocol in each shard has been proved to be secure, i.e., it meets the definition of A Secure Sharding Blockchain (see Appendix A for a detailed definition).*

**Assumption A2.** *We assume that each shard in the blockchain network maintains a majority of honest nodes, i.e., a proportion for each shard.*

For the basic assumptions of the security analysis, we propose:

**Lemma A1.** *Without cross-shard TX, every shard can achieve security.*

**Proof of Lemma A1.** Based on Assumptions A1 and A2, each shard has honest majority of nodes. The security aspects of Persistence, Liveness, and Consistency depend on the parameters $\mu$, network condition, and k, which has been satisfied by Assumption A1. Even if $Q_i$ is large, it will not affect the network condition. The safety coefficient k is met due to $\mu = 1 - a$ ([4] Theorem 1) and by the consensus algorithm within the shard. Based on our assumption that invalid relay transactions will not affect other shards, each shard can run independently, thus ensuring its security. $\square$

**Theorem A1.** *Even in the worst case, as long as* $P_i U < \frac{1}{h}$ *is satisfied, persistency and liveness can be guaranteed with a very high probability.*

**Proof of Theorem A1.** Persistency depends on two factors: the probability that stable transactions become invalid and the probability that confirmed cross-shard transactions are revoked. These two factors only depend on the common prefix property of the shard consensus mechanism. Based on Definition A2, the blockchain protocol has been proof atomic for cross-shard TX that invalidating relay TX will not affect other shards, thus satisfying the common prefix property. It is assumed that the common prefix is satisfied with a probability of $1 - p$ (overwhelmingly in the "depth" security parameter k). The p value is a very small probability value under a normal operating blockchain protocol. We know that there may be less than U shards involved in a transaction. Therefore, the probability that a single cross-shard transaction is valid is bigger than $(1 - p)^U$. Here, we assume that U is a constant, and in practical scenarios, each transaction typically involves a limited number of inputs and outputs, or UTXOs. Consequently, we can regard U as a constant. In the worst case, all transactions in $P_i$ are cross-shard transactions, which follow the binomial distribution $B\left(\left|P_i\right|, (1 - p)^U\right)$; we can get the probability that all transactions in $P_i(t)$ are valid: $\text{Prob}(x = |P_i|) = (1 - p)^{|P_i|U}$. To find $1 - \text{Prob}(x = P_i) < \varepsilon$, consider its Taylor expansion: $\left(1 - x\right)^n = 1 - nx + \frac{n(n-1)}{2}x^2 - \ldots$

$$(1 - p)^{P_i U} \approx 1 - P_i U p + \frac{P_i U(P_i U - 1)}{2} p^2$$

Then the probability that the transaction is invalidated is:

$$1 - \text{Prob}(x = P_i) \approx P_i U p - \frac{P_i U(P_i U - 1)}{2} p^2 + \frac{P_i U(P_i U - 1)(P_i U - 2)}{6} p^3$$

Consider the following inequality:

$$P_iUp - \frac{P_iU(P_iU - 1)}{2}p^2 + \frac{P_iU(P_iU - 1)(P_iU - 2)}{6}p^3 < \epsilon$$

where p satisfies the condition $0 < p < h\epsilon$, c is a known positive constant, and $h\epsilon$ is the probability of error that the blockchain protocol can tolerate. Our goal is to determine an upper bound on n such that the above inequality holds for all p and $\epsilon$ values that satisfy the given conditions.

By substituting $p = h\epsilon$ and simplifying, we get:

$$P_iUh\epsilon - \frac{P_iU(P_iU - 1)}{2}(h\epsilon)^2 + \frac{P_iU(P_iU - 1)(P_iU - 2)}{6}(h\epsilon)^3 < \epsilon$$

Moreover, $\epsilon$ is a positive number that can take any value close to zero. we can conclude the upper bound: $P_iU < \frac{1}{h}$.

Therefore, to ensure that the above inequality holds for all p and $\epsilon$ values that meet the conditions, $P_iU$ must be less than $\frac{1}{Uh}$. From the above corollary, we can know that persistency is satisfied. Similarly, liveness is satisfied within each shard. Moreover, this means that cross-shard transactions also satisfy liveness. Specifically, as long as the chain quality and chain growth are ensured within each shard, both active and relay transactions will eventually be included in the shard transaction ledger. □

**Theorem A2.** *As long as the cross-shard protocol of the verified sharded blockchain protocol satisfies atomicity and* $P_iU < \frac{1}{h}$, *for* $\forall i \in M$, *then the blockchain protocol satisfies consistency*

**Proof of Theorem A2.** It is known that, if the cross-shard protocol satisfies atomicity, then the adversary cannot validate two conflicting transactions across different shards. When $P_iU < \frac{1}{h}$, the liveness and persistence of all shards can be guaranteed by Theorem A1, and a cross-shard transaction is "stable" with probability $1 - p^U$ when all associated shards are accepted. Therefore, the adversary cannot revert the chain of a shard and double-spend an input of the cross-shard transaction because consistency holds with high probability, given persistence holds with high probability. □

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
