# Peer review of "Solving Confirmation Time in Sharded Blockchain with PFQN"

_electronics, doi:10.3390/electronics13071220_

Round 1
Reviewer 1 Report
Comments and Suggestions for Authors
The authors present a Product-Form Queue Network (PFQN) model to address the problem of transaction confirmation time in sharded blockchains. They also incorporate a new confirmation queue to more accurately simulate the actual process of confirming transactions in the blockchain. Finally, they perform a detailed quantitative analysis of the relationship between network load, consensus efficiency and security in sharded blockchains to make the sharded blockchain more robust and efficient.
To improve the work done so far, I suggest the following additions to the document:
1) Introduction: The authors should add a few more references, for example when they cite sharding paradigms (Line 29) and representative sharding solutions (Line 36). They could also state why they chose to use a PFQN queue and whether there are other alternative methods to use (possibly including a comparative table of merits and demerits).
2) References: The references are not ordered and some are not shown in the text, such as numbers 5, 6, and 11.
3) Related Works: It would be useful to include a comparative table of queue models with their strengths and weaknesses.
4) Lines 111-119: it is not clear what is meant, perhaps the text is incomplete?
Reviewer 2 Report
Comments and Suggestions for Authors
The authors investigated the transaction time in blockchain. To address the issue, the authors implemented Product Form Queue Network (PFQN) model.
Although the authors claim the potential speedup in the confirmation time, I have serious concerns in the security of blockchain.
As widely known in this field, the security of the conventional blockchain is not guaranteed due to the quantum algorithms. See the following textbook
https://www.sciencedirect.com/science/article/abs/pii/S0065245818300160
(see https://epubs.siam.org/doi/10.1137/S0097539795293172 also)
How to overcome this problem? The authors must elaborate more on the security, since it is the most crucial point of the P2P database.
Can the PFQN model be useful to quantum digital signature as well? If so, the author could solve the problem at a significant level. For example, the authors should explain the possibility for the following quantum-secure blockchains:
https://www.nature.com/articles/s41598-022-12412-0
https://link.springer.com/chapter/10.1007/978-3-030-01174-1_58
https://www.mdpi.com/2227-7390/11/18/3947
https://www.worldscientific.com/doi/10.1142/9781786348210_0006
Round 2
Reviewer 2 Report
Comments and Suggestions for Authors
The authors submitted a revised manuscript, however nothing was improved and the authors did not solve the previous concern raised in the previous report.
What I am saying is all the authors are doing here is not resistant against very standard quantum attacks. This is widely known among the blockchain community, however the authors completely neglect this weakness.
They must address this significant issue, otherwise the paper is not publishable.
Read the previous report more carefully and reflect the concerns into their manuscript.
Author Response
Dear Reviewer,
Thank you for your valuable feedback and for pointing out the critical issue of quantum security in blockchain systems. We acknowledge the importance of addressing quantum vulnerabilities, especially with the advancement of quantum computing technologies. In response to your concerns, we have expanded our discussion on the security measures against quantum attacks and provided additional clarity and evidence to support our approach. Here's a summary of the revisions and additions made to the manuscript:
-
Expanded Explanation on Quantum Security in Blockchain: We have elaborated on the potential threats posed by quantum computing to blockchain security, specifically regarding the conventional cryptographic algorithms currently employed. We recognize the significance of this issue and have included a more detailed analysis in Section 4.2.
-
Integration of Quantum-Resistant Mechanisms: In response to the concerns raised, we have further detailed how our PFQN model incorporates quantum-resistant mechanisms. We've clarified how assuming quantum-resistant digital signature and encryption algorithms can be implemented in a sharded blockchain allows our model to simulate the impact of these quantum-resistant measures on transaction confirmation times. We believe this addition addresses your concern about the need for a more concrete approach to quantum security.
-
References to Current Research: We have updated our literature review to include additional recent studies that focus on quantum-safe blockchain technologies. This includes the integration of concepts such as Quantum Key Distribution (QKD) and lattice-based encryption, which are deemed to be resistant against quantum attacks. We have provided a clearer connection between these technologies and their applicability to the PFQN model to ensure that our approach aligns with current quantum-safe methodologies.
-
Detailed Attack Models and Assumptions: Based on your feedback, we have expanded our discussion on attack models and assumptions to include potential quantum attack vectors, such as the Grover and Shor algorithms. We have described in greater detail how these attacks could impact blockchain components and outlined specific measures that can be taken to mitigate such risks.
-
Empirical Analysis and Theoretical Backing: We have included additional empirical data and theoretical analysis to support our assertions regarding the security of the PFQN model under quantum threats. This includes an extended discussion on the computational complexity of post-quantum cryptographic algorithms and their integration into the blockchain framework.
-
Direct Responses to the Provided References: We have carefully reviewed the provided references regarding quantum-secure blockchains and incorporated insights and findings from these studies into our revised manuscript. This is reflected in our expanded discussion in sections 4.2.1 and 4.2.2, where we compare our approach with the proposed quantum-safe solutions.
We hope that these revisions adequately address your concerns regarding the quantum security of blockchain technologies as discussed in our paper. We have strived to provide a comprehensive response to your feedback, underlining the potential of the PFQN model in evaluating and enhancing the quantum resilience of blockchain systems.
Thank you again for your constructive feedback, which has significantly contributed to improving the quality and depth of our research.
Sincerely,
Junting Wu
Round 3
Reviewer 2 Report
Comments and Suggestions for Authors
Now the paper has been improved significantly, although it can be better. I do hope the authors address the open problems in their future works, but in the meanwhile the paper should be discussed more openly including more experts of the secure blockchain transactions.